# Spin-photon entanglement with direct photon emission in the telecom C-band

P. Laccotripes [1,2] ✉, T. Müller [1] ✉, R. M. Stevenson[1], J. Skiba-Szymanska[1], D. A. Ritchie [2] & A. J. Shields[1]

Quantum networks, relying on the distribution of quantum entanglement between remote locations, have the potential to transform quantum computation and secure long-distance quantum communication. However, a fundamental ingredient for fibre-based implementations of such networks, namely entanglement between a single spin and a photon directly emitted at telecom wavelengths, has been unattainable so far. Here, we use a negatively charged exciton in an InAs/InP quantum dot to implement an optically active spin qubit taking advantage of the lowest-loss transmission window, the telecom C-band. We investigate the coherent interactions of the spin-qubit system under resonant excitation, demonstrating high fidelity spin initialisation and coherent control using picosecond pulses. We further use these tools to measure the coherence of a single, undisturbed electron spin in our system. Finally, we demonstrate spin-photon entanglement in a solid-state system with entanglement fidelity $F = 80.07 \pm 2.9\%$, more than 10 standard deviations above the classical limit.

Simple quantum networking functionalities, such as quantum key distribution between not-too-distant points, are already a reality, and more sophisticated schemes, such as blind quantum computing, clock synchronisation, and entanglement distribution between any points on the globe, are currently in development[1]. A convenient way of transporting quantum information over long distances is via photons. For fibre-based quantum networks, photon wavelength in the telecom C-band is imperative for long-distance transmission due to the low-absorption window ~1550 nm.

Solid-state-based spin-photon interfaces[2] are a practical and integrable resource for long-distance quantum networking. In particular, they are indispensable for many quantum repeater schemes, whether reliant on quantum memories[3] or the utilisation of complex multiphoton entangled states[4–8]. Many important building blocks have already been demonstrated, such as entanglement between photons and stationary qubits[9–12], entanglement between separated solid-state spins[13,14], and the generation of multiphoton entanglement[15–21]. However, their practical range of applications is limited because of their emission wavelengths in the visible range of the spectrum up to 900 nm.

Wavelength conversion is one option to circumvent this problem[11,22,23]. While it can be beneficial for applications requiring long coherence times, such as quantum memories[24], the increased experimental complexity and inherent losses can significantly hinder the efficiency of entanglement distribution and render applications such as multiphoton entangled state generation unfeasible. Alternatively, a few systems have recently emerged that allow for direct emission into the telecom C-band. Most prominently, impressive progress has been made using quantum dots (QDs) based on strain-relaxed InAs/GaAs or InAs/InP, culminating in the recent demonstrations of coherent spin manipulation[25] in the strain-relaxed InAs/GaAs system, and photon emission approaching the Fourier-limit in the InAs/InP system[26], respectively. However, the crucial ingredient for the above applications, namely entanglement between a single spin and a photon at telecom wavelengths, has been elusive so far.

Here, we significantly narrow the performance gap between telecom-wavelength QDs and the more established solid-state systems with emission at shorter wavelengths. We transfer the established methods for manipulation of a solid-state spin[27–30] to control the spin

[1]Toshiba Europe Limited, 208 Science Park, Milton Road, Cambridge CB4 0GZ, UK. [2]Cavendish Laboratory, University of Cambridge, JJ Thomson Avenue, Cambridge CB3 0HE, UK. ✉e-mail: pl490@cam.ac.uk; tina.muller@toshiba.eu

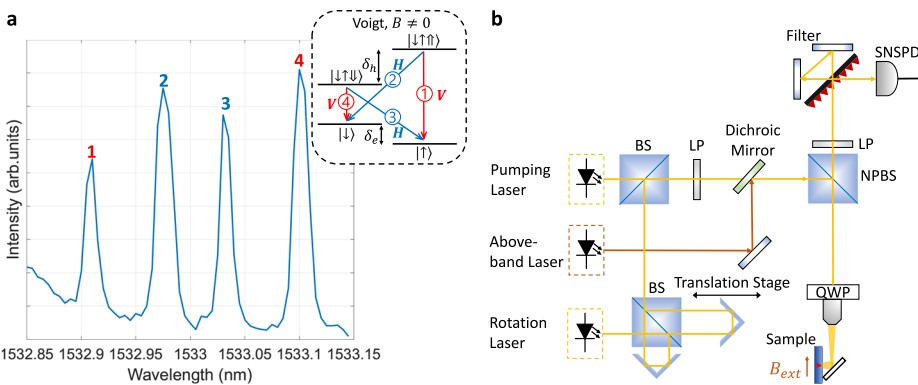

**Fig. 1 | QD spin-qubit system and experimental setup. a** Photoluminescence absorption spectrum at 5 T magnetic field in Voigt geometry. Inset: Trion level structure with radiative transition selection rules. **b** Schematic of the experimental setup; BS beam splitter, LP linear polariser, QWP quarter-wave plate, NPBS non-polarising beam splitter, SPCM single-photon counting module, SNSPD superconducting nanowire single-photon detector.

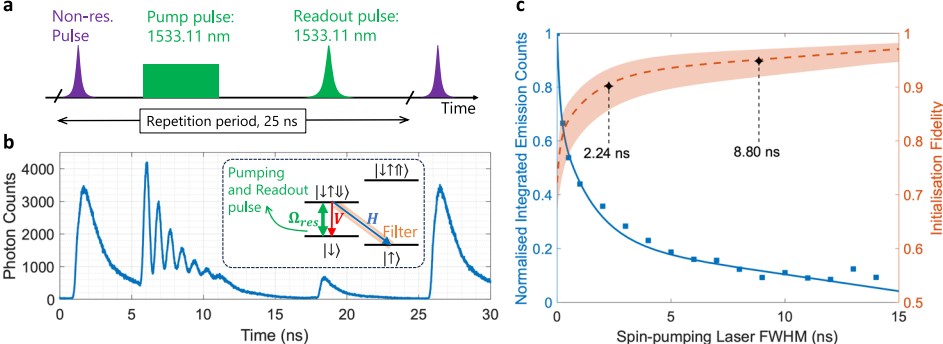

**Fig. 2 | Initialisation of the spin-qubit state. a** Excitation pulse sequence, as described in the main text. **b** Time-resolved measurement of the $|{\downarrow}{\uparrow}{\Downarrow}\rangle - |{\uparrow}\rangle$ transition. Emission during the resonant pump pulse exhibits Rabi oscillations with a period 0.864 ns (1.158 GHz). Inset: Trion level schemes with relevant trasnitions. **c** Normalised integrated emission intensity of the readout pulse (blue) as a function of spin-pumping laser duration. The blue data points are fitted with a rate equation model to extract the evolution dynamics of the $|{\uparrow}\rangle$ state and hence the spin initialisation fidelity $F_{\text{init}}$ (orange). Poissonian errors for the measured data points are contained within the square markers, and for the spin initialisation fidelity the 95% CI is illustrated by the shaded region.

in an InAs/InP QD, demonstrating spin initialisation and coherent spin rotations. We further use these tools to measure the coherence of a single, undisturbed electron spin in our system. Finally, we demonstrate entanglement between the electron spin and the frequency of a photon at C-band telecom wavelengths.

## Results

In this work, we utilise a spin-qubit system, based on a single electron, which is optically injected into a QD using a non-resonant laser pulse. Resonance fluorescence measurements are used to map the optical transitions of our dot (see Supplementary Information 2 and 3). We obtain the absorption spectrum shown in Fig. 1a by scanning the resonant laser across the four allowed transitions. Under the influence of an external magnetic field in Voigt geometry (i.e., applied perpendicularly to the direction of growth), the degeneracy of eigenstates is lifted, and a double $\Lambda-$system is formed from the initial two-level system, due to the Zeeman effect. All four allowed transitions between the ground and excited states are linearly polarised either parallel (H) or perpendicular (V).

As the magnetic field is increased, a clear fourfold line splitting is observed. At 5 T, there are four distinct optically addressable transitions, Fig. 1a. The transitions are labelled 1–4 and correspond to the optical selection rules as illustrated in the inset of the figure. By fitting the emission peaks with a Lorentzian function, a trion-excited and electron-ground energy Zeeman splitting of $\delta_h = 65.3 \pm 0.5\,\mu\text{eV}$ ($15.8 \pm 0.1\,\text{GHz}$) and $\delta_e = 35.3 \pm 0.5\,\mu\text{eV}$ ($8.5 \pm 0.1\,\text{GHz}$) is obtained, respectively.

Figure 1b illustrates the experimental setup that allows us to implement above-band and resonant laser excitation as well as selective detection of resonance fluorescence from each of the four allowed transitions (see Supplementary Information 2). The sample is kept at a temperature of ~4K in the Voigt configuration. Pulsed, resonant excitation is achieved using a CW 1550 nm laser in conjunction with an EOM that is driven by an arbitrary wavefunction generator (AWG). Spin control experiments are conducted using a ps-pulsed rotation laser along with a Ramsey interferometer. The fluorescence light is collected and filtered by a confocal microscope setup which removes the resonant laser contribution via cross-polarisation filtering. A free-space grating-based filter is used to spatially separate emissions from individual excitonic complexes before detection.

### Coherent spin manipulation

In this section, we demonstrate the spin control tools required for the spin-photon entanglement experiment. We start by initialising the spin into the desired ground state $|{\uparrow}\rangle$ via optical pumping[31]. In the spin-initialisation protocol adopted here, a three-pulse sequence is used at a 40 MHz repetition rate, as shown in Fig. 2a. Initially, a non-resonant pulse injects carriers into the QD, forming a trion state with a random hole spin, and serving as a spin-reset mechanism at the start of each cycle. The trion then recombines radiatively with equal probability into one of the two electron-ground states. Next, a $\sigma^+$-polarised square pulse of varying FWHM and resonant with the $|{\downarrow}\rangle - |{\downarrow}{\uparrow}{\Downarrow}\rangle$ transition, is applied 5 ns after the charge injection pulse, to achieve the optical

pumping. After the pump pulse, optionally a spin rotation pulse is applied for coherent spin rotations (see below). Finally, the population of the $|\downarrow\rangle$ is probed by applying another $\sigma^+$-polarised readout pulse resonant with $|\downarrow\rangle - |\downarrow\uparrow\Downarrow\rangle$ transition, 20 ns after the pump pulse. This is a Gaussian pulse with a FWHM of 0.3 ns.

The resonantly driven transition is shown in the inset of Fig. 2b (green arrow). Spectral filtering of the diagonal transition in the $\Lambda$ system allows us to trace the QD's emission dynamics, with the time-resolved fluorescence shown in Fig. 2b. The non-resonant pulse results in an exponentially decaying emission trace with the decay time given by the excited state lifetime, $\Gamma = 1.32$ ns. During the pump pulse, an oscillatory pattern is observed, which is evidence of the coherent interaction between the pump laser and the QD system. During each emission cycle, there is a 50% probability of the system decaying to the $|\downarrow\rangle$ state via emission of a $V$-polarised photon, in which case it will be re-excited, or to the $|\uparrow\rangle$ state, via emission of an $H$-polarised photon, in which case no further excitation occurs. The probability of successful initialisation to the $|\uparrow\rangle$ state scales with $P_{init} = 1 - 0.5^{(n+1)}$, where $n$ is the number of cycles the system was excited and allowed to relax. Therefore an increase in state preparation fidelity with longer pump pulse duration is expected, which

will manifest itself in a suppression of the integrated emission after the readout pulse.

In Fig. 2c, the integrated emission counts after the probe pulse are plotted as a function of pump pulse duration, and normalised to the counts observed in the absence of pumping. A clear suppression is observed with increasing pump pulse duration. To extract the state preparation fidelity, a theoretical rate equation model is developed following ref. 25 (see Supplementary Information 4). To fit the experimental data, the integrated emission intensity is calculated from the theoretical model. The fidelity is, in turn, determined by the state population given by the model. A spin preparation fidelity of 90 % (with 95 % CI [85.9, 93.3]) and 95 % (with 95 % CI [92.2, 97.0]) is achieved after only 2.2 ns and 8.8 ns of spin-pumping time, respectively. We also observe a preparation fidelity of the $|\uparrow\rangle$ state exceeding 97.1% (with 95 % CI [94.9, 98.2]) for the longest pumping window. These values could be further improved by increasing the power and duration of the pump pulse.

Next, we demonstrate optical spin rotations using a detuned laser pulse with a few ps duration, inducing spin rotations around the $x$ axis on the Bloch sphere as indicated in Fig. 3b. We employ a circularly polarised pulse to ensure that the probability amplitudes of the two

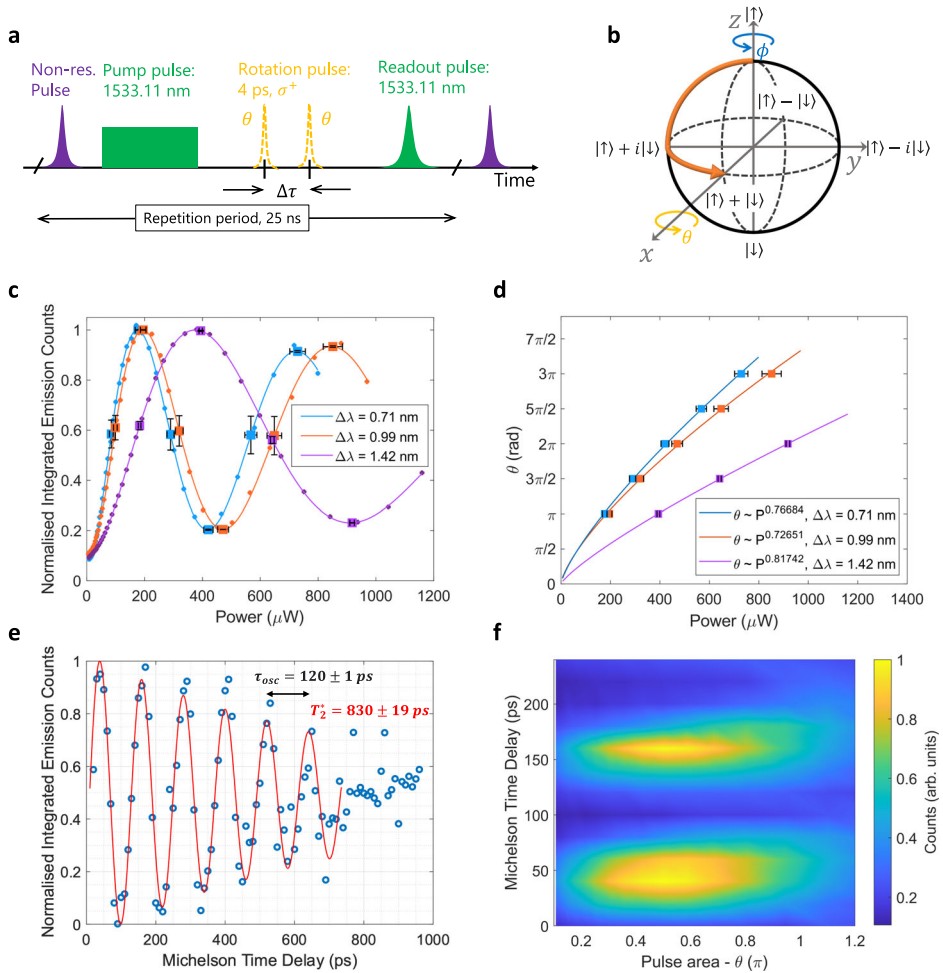

**Fig. 3 | Spin rotation and full coherent control. a** Excitation pulsed sequence. A non-resonant pulse first injects carriers into the QD followed by a pump pulse which initialises the spin into $|\uparrow\rangle$. Two circularly polarised ps-pulses with time delay $\Delta\tau$ rotate the electron's spin. A resonant readout pulse is used to probe any population remaining in the $|\downarrow\rangle$ state. **b** Pictorial representation of the electron qubit evolution along the Bloch sphere. **c** Integrated emission intensity as a function of a single rotation pulse power, at 9 T, for three different red-detunings from the lowest energy transition, demonstrating coherent control of the electron's spin.

The data are fitted to extract the power values at angles $\theta = m\pi/2 \; \forall m \in 1, 2, 3, 4, 5, 6$, as marked by the error-barred squares (95% confidence interval). **d** Rotation angle versus pulse power fit results in an average power dependence of $\theta \propto P_{rot}^{0.77}$ for the three detunings. **e** Ramsey interference for a pair of rotation pulses, at 5 T, red detuned by $\Delta\lambda = 0.6$ nm from the lowest energy transition, showing normalised integrated emission counts as a function of the time delay between the pulses. **f** Full coherent control of the spin-qubit at 5 T. Normalised integrated emission counts as a function of rotation angle $\theta$ and time delay between the rotation pulses.

transitions in the effective $\Lambda$ system add constructively, and large red-detunings, from the lowest energy transition, to minimise undesired incoherent excitation[27,32] (see Supplementary Information 5). In this case, the rotation angle $\theta$ is directly proportional to the rotation laser pulse power and inversely proportional to the detuning $\Delta\lambda$. For detunings larger than the electron-ground state splitting, the net effect is that the spin state population oscillates with an effective Rabi frequency, $\Omega_{Reff}$[27]. Here, we chose a magnetic field of 9 T for a ground-state splitting of 16 GHz (66 µeV). We employ the same pulse sequence as shown in Fig. 2a, for a fixed 6-ns pumping time. In addition, we now insert the rotation pulses (dotted-yellow pulses) 9 ns after the beginning of the pulse sequence, Fig. 3a. The final population of the $|\downarrow\rangle$ state is probed using a third resonant $\sigma^+$-polarised Gaussian pulse as before. The recorded intensity data are fitted using an exponentially decaying sinusoid, allowing us to extract the power values at which $\theta = m\pi/2 \ \forall m \in 1, 2, 3, 4, 5, 6$. These points are marked in Fig. 3c with square markers, and further plotted in Fig. 3d for all three detunings. The achieved rotation angles as a function of rotation power can be fitted using a power law, and we empirically determine that $\theta \propto P_{rot}^{0.77}$ in the range $\pi \geq \Theta \geq 3\pi$. While a linear dependence is expected for the ideal STIRAP protocol[33], the sub-linear dependence observed here is an expression of the breakdown of the adiabatic elimination of the excited states in the rotation process[27]. This means that there is a finite probability of finding the system in the excited state, from which radiative decay can take place.

In Fig. 3c, we plot the integrated emission counts after the readout pulse as a function of the rotation pulse power for the three different detunings $\Delta\lambda = [0.71 \text{ nm}, 0.99 \text{ nm}, 1.42 \text{ nm}]$. For all three detunings, clear oscillations can be seen related to coherent Rabi rotations of the spin state population. We observe $\theta$ exceeding $3\pi$, only limited by the available laser power. As expected, the larger the detuning, the higher the laser power required to achieve the same rotation $\theta$. We observe some amplitude damping over the measured power range, which is most likely related to a combination of dephasing processes, i.e., decrease of the Bloch vector length as $\theta$ increases, and imperfect alignment of the rotation vector due to the QD asymmetry and non-ideal circular polarisation of the rotation pulses. Moreover, due to the finite fidelity of spin initialisation (~93% for 6 ns of optical pumping), the integrated emission counts do not start at zero in the limit of 0 µW rotation pulse power.

Rabi oscillations demonstrate the rotation of a qubit by a single axis, $U(1)$ control. Full coherent control, $SU(2)$ control, is achieved by rotation about a second axis, $\phi$, which can be realised by utilising the inherent Larmor precession in the applied magnetic field. Here, we chose a magnetic field of 5 T for a Larmor precession of ~120 ps. We employ detuned circularly polarised ps-pulses and make use of the precession to achieve rotation by $\theta$ and $\phi$, respectively. The spin state is probed following excitation by two $\theta$-pulses separated by a time delay $\Delta\tau$ set by the Ramsey interferometer, Fig. 3a. Ramsey fringes are observed in the integrated counts of the readout pulse, Fig. 3e. From the Ramsey fringes we extract an inhomogeneous dephasing time $T_2^* = 830 \pm 19$ ps. Due to the temporal modulation of the pumping laser the $T_2^*$ is significantly higher compared to CW pumping schemes[25] and can be significantly improved using spin refocusing techniques[34].

Finally to access any arbitrary state on the Bloch sphere, we vary the rotation pulse power and time delay between the two pulses simultaneously. Figure 3f shows the normalised integrated emission counts as a function of rotation pulse area, $\theta(\pi)$, and time delay between the pulses, $\Delta\tau$. As $\theta$ increases from 0 to $\pi$ the amplitude of the Ramsey fringe first increases achieving a maximum at $\pi/2$ before decreasing again. High fidelity $\pi/2$ pulses are of significant interest as they are used in the following section to measure quantum correlations in the superposition basis. By considering the visibility of the Ramsey fringes, we estimate the fidelity of our $\pi/2$ pulses to be $F_{\pi/2} = 93 \pm 0.7$ % (see Supplementary Information 5).

## Spin-photon entanglement in the computational basis

Having demonstrated all the necessary tools required for coherent control and spin-photon entanglement in InAs/InP QDs emitting at telecom wavelengths, we proceed to measure entanglement between the frequency of an emitted photon and the resident spin in the QD. Following excitation of the QD into the trion state, radiative decay projects the spin and photonic qubit system into the following entangled state:

$$|\psi\rangle = \frac{1}{\sqrt{2}}(|\omega_{red}, \downarrow\rangle e^{-i\omega_z(t - t_{phot})} + i|\omega_{blue}, \uparrow\rangle) \qquad (1)$$

where $t_{phot}$ is the photon generation time and the Zeeman splitting $\omega_z = \omega_{blue} - \omega_{red}$. This process is schematically shown in Fig. 4a. We first measure correlations in the computational basis. In this case, the relative phase between the two components of the entangled state is not important, simplifying to a measurement of $|\omega_{red}, \downarrow\rangle$ and $|\omega_{blue}, \uparrow\rangle$.

We evaluate our scheme for two separate magnetic field strengths in Voigt geometry. We choose 5 T, resulting in a ground-state spin splitting of ~0.065 nm, for optimal overall entanglement fidelity, and 9 T, resulting in a ground-state spin splitting of ~0.125 nm, for highest correlation contrast in the computational basis. A sufficiently high magnetic field further ensures adequate separation between the trion energy levels, $|\downarrow\uparrow\Downarrow\rangle$ and $|\downarrow\uparrow\Uparrow\rangle$, and consequently efficient resonant excitation.

To verify the single-photon nature of the emission, and to determine the optimal time window over which to evaluate the correlations, we first perform a Hanbury-Brown and Twiss measurement. Figure 4c illustrates the second-order autocorrelation function, $g^{(2)}(\Delta\tau)$, evaluated in the time window [1.15 ns, 1.7 ns] of each measurement cycle, as well as for the whole entanglement pulse [1 ns, 4 ns]. The excitation pulse arrives at ~800 ps with a duration of 300 ps. By time-filtering we avoid any contributions from the scattered laser photons, as well as the detector's dark count rates. The observed value of $g^{(2)}(0) \approx 0.02 \pm 0.02$ in the narrow time window clearly indicates photon antibunching and limited multiphoton emission, which can lead to entanglement degradation. This is in stark contrast to the value of $g^{(2)}(0) \approx 0.20 \pm 0.02$ for the whole pulse. Thus, to limit the influence of experimental artefacts in spin-photon correlations, we will evaluate our measurements in the time window [1.15 ns, 1.7 ns].

Demonstrating spin-photon correlations in the computational basis requires measuring classical correlations between the electron's spin and a photon degree of freedom. Spontaneous emission from the trion state at a frequency $\omega_{red}$ or $\omega_{blue}$ leaves the resident electron in the $|\downarrow\rangle$ or $|\uparrow\rangle$ state, respectively. As the photon polarisation information is erased in our resonant excitation scheme, we measure photon frequency, $|\omega_{red}, \downarrow\rangle$ and $|\omega_{blue}, \uparrow\rangle$ using a free space grating-based filter with a linewidth of $\Delta\lambda = 0.12$ nm. The pulsed excitation scheme used for the correlation measurements is depicted in Fig. 4b. First, the dot is injected with charges using a non-resonant pulse and the system is initialised into a random $|\downarrow\rangle$ or $|\uparrow\rangle$ state. Then, a 300-ps Gaussian-shaped entanglement pulse tuned to the transition $|\downarrow\rangle - |\downarrow\uparrow\Downarrow\rangle$ excites the system to the trion state, from which an entangled spin-photon pair is generated. A readout pulse follows 9 ns after. A photon detected in a time window following the readout pulse indicates with very high confidence that the spin before the measurement was in the $|\downarrow\rangle$ state. Conversely, the absence of any detection event yields no information. Hence, to project the spin-qubit into the $|\uparrow\rangle$ state, an additional $\pi$-pulse is applied (Fig. 4b dashed yellow line) 4 ns after the onset of the entanglement pulse.

The entanglement fidelity information in the computational basis is extracted by measuring coincidences between resonance fluorescence photons at $\omega_{red}$ or $\omega_{blue}$ from the entanglement pulse, conditioned on the detection of a photon during the subsequent readout

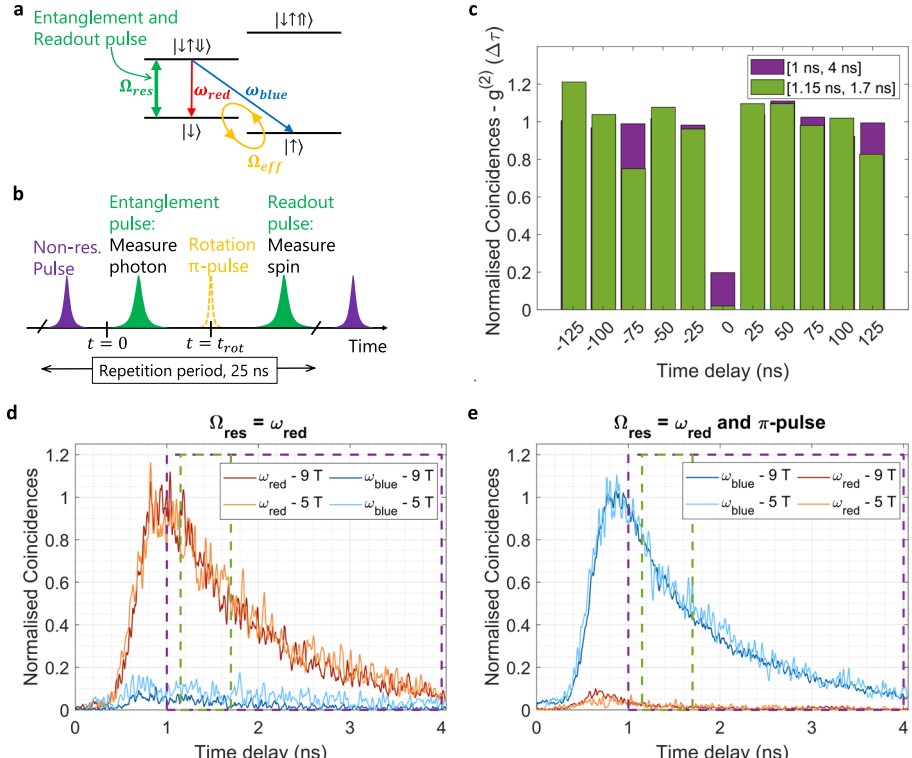

**Fig. 4 | Measurement of classical spin-photon correlations. a** Level scheme showing the resonant entanglement/readout laser, the (optional) rotation pulse and the blue and red decay paths. **b** Pulse sequence used for the entanglement measurements. **c** Second-order autocorrelation measurement to evaluate the purity of our source. Two different time windows, [1.15 ns, 1.7 ns] marked in green, and [1 ns, 4 ns] marked in purple, in **d**, **e**, are considered which correspond to part or all of entanglement pulse decay, respectively. **d** Measured arrival times of red/blue photons after the entanglement pulse, conditioned on a photon being present in the readout pulse (spin projection to the $|\downarrow\rangle$ state). Measurements were carried out at 9 T (dark red and dark blue curves) and 5 T (light red and light blue curves). **e** The same measurement as in **d**, but with a $\pi$ pulse inserted before the readout pulse. The readout, therefore, projects the spin into the $|\uparrow\rangle$ state.

pulse. If a photon was detected during the readout pulse, the electron's spin is projected to $|\downarrow\rangle$ after the entanglement pulse. Figure 4d shows normalised coincidences for the pulse sequence at 5 T and 9 T, conditioned on the detection of the spin state $|\downarrow\rangle$. There is a clear suppression of $\omega_{blue}$ photons compared to $\omega_{red}$ photons with a ratio of ~16:1 (9 T) and ~7:1 (5 T) in the [1.15 ns, 1.7 ns] interval, as expected from the ideal projection of the entangled state to $|\omega_{red}, \downarrow\rangle$. The better ratio observed at 9 T is entirely due to the better performance of our frequency filter at higher splittings (see Supplementary Information 6). To completely characterise spin-photon correlations in the computational basis, we introduce a $\pi$-pulse that allows us to measure coincidences conditioned on the $|\uparrow\rangle$ spin state. In Fig. 4e, the ratio of blue to red photons is ~29:1 (9 T) and ~16:1 (5 T). The ratios of blue to red photons are higher due to some unwanted re-excitation by the $\pi$-pulse and not as optimal filtering of the transition $\omega_{blue}$ compared to $\omega_{red}$. From the above measurements the fidelity $F_1$ in the computational basis is $F_1^{9T} = 92.87 \pm 1.2\%$ and $F_1^{5T} = 84.13 \pm 1.1\%$ for 9 T and 5 T, respectively (See Supplementary Information 7).

**Spin-photon entanglement in superposition basis**
To quantify the spin-photon entanglement in our system, it is necessary to demonstrate spin-photon correlations in three mutually orthogonal bases. In addition to the measurements in the computational basis, we now consider correlations in two orthogonal superposition bases. Now, the relative phase of the two components in Equation (1) can no longer be neglected, since the red and blue frequency components have different propagation phase factors. The resident electron's spin is rotated using a $\pi/2$ or a $3\pi/2$ pulse at time $t_{rot}$ ~ 4 ns. A measurement of a photon during the readout pulse projects the spin to $|\downarrow\rangle$ at $t_{meas}$. Thus, at the onset of the rotation pulse $t_{rot}$,

the system is projected into a superposition state with orthogonal phase for the two rotation pulses considered, $|\uparrow\rangle \pm i|\downarrow\rangle$[10]. The time evolution can then be traced backwards where oscillations in the spin populations can be observed. In the spin-photon correlations measurement, these oscillations will manifest at the beating frequency of $\omega_{blue}$ and $\omega_{red}$, $\omega_z$. Due to the random emission time of photons, $t_{phot}$ and the fact that the SSPDs jitter is $<1/\omega_z$, these oscillations are sampled, effectively implementing a projective measurement onto the orthogonal states $|\omega_{blue}\rangle \pm |\omega_{red}\rangle$[10].

For the subsequent correlation measurements, a broader optical filter with a bandwidth of ~0.5 nm, encompassing both $\omega_{blue}$ and $\omega_{red}$, was used. Again, correlations were evaluated at magnetic fields of 5 T and 9 T. Figure 5 shows coincidences between a photon detected during the readout and entanglement pulses. In Fig. 5b, d, the coincidences are measured following a $\pi/2$ pulse, while in Fig. 5c, e following a $3\pi/2$ pulse. The oscillations have a period of $2\pi/\omega_z$ of $110 \pm 1.5$ ps at 5 T, and $66 \pm 1.2$ ps at 9 T. While clear oscillations are visible in the raw data, the period of 66 ps at 9 T approaches the timing jitter of our detector system (40 ps) and hence oscillation amplitudes are reduced along with large error bars. Further technical limitations include variations in the effective rotation laser power due to drifts in the microscope alignment over the measurement duration, as well as imperfect calibration of the circular detection basis. We suspect these limitations to have been the root cause for the mismatch in the oscillation amplitude between Fig. 5b, c.

The normalised coincidence events are then fitted and deconvolved with the 40-ps Gaussian response representing the detector's resolution. At 9 T magnetic field we measure a rotated basis fidelity of $F_2^{9T} = 58.85 \pm 7.4\%$ and at 5 T $F_2^{5T} = 76.01(4.6)\%$, extracted from the visibility of the normalised coincidences (See supplementary

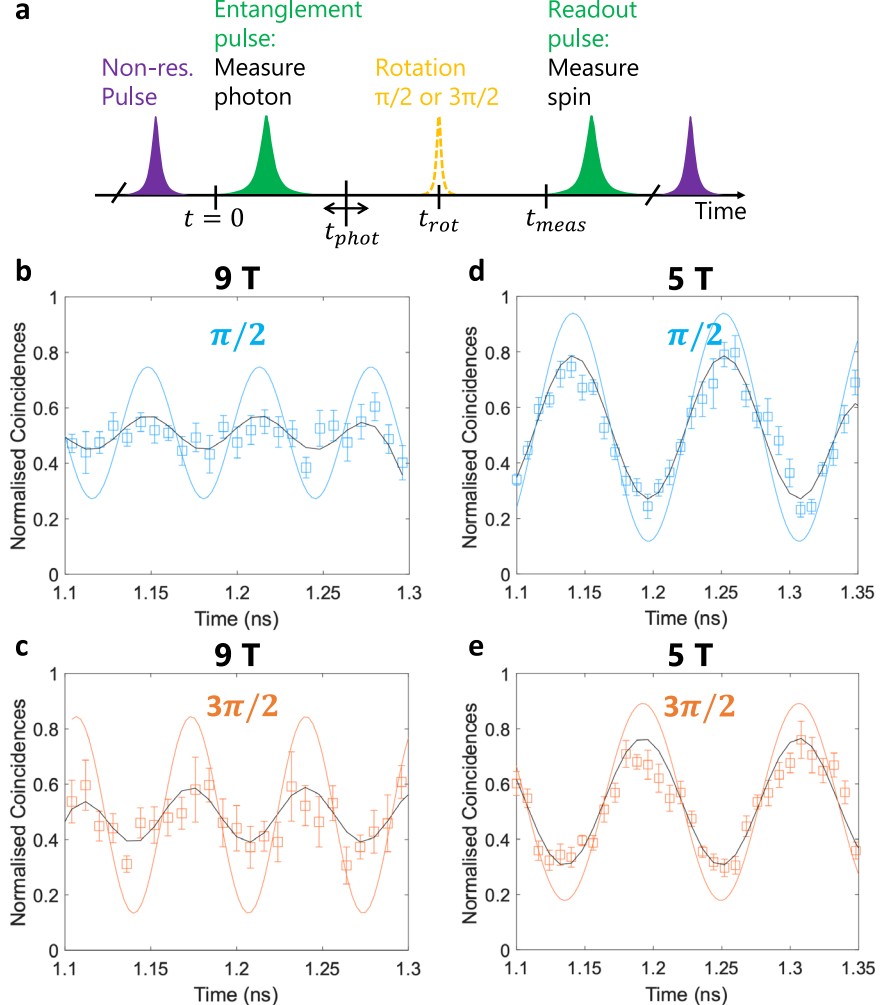

**Fig. 5 | Measurement of quantum spin-photon correlations. a** Schematic of the pulse sequence used for the correlation measurements in superposition bases. **b, c** Time-resolved coincidence events when the spin state was projected onto $|\uparrow\rangle + i|\downarrow\rangle$ and $|\uparrow\rangle - i|\downarrow\rangle$, respectively, recorded at a magnetic field of 9 T. **d, e** identical measurements to **b, c**, recorded at a magnetic field of 5 T. Black lines correspond to fits of the original data which are marked by squares with error bars given by poissonian statistics, and coloured lines are after deconvolving with the detectors' jitter.

Information 7). It also evident that spin detection events along $(|\downarrow\rangle + i|\uparrow\rangle)/\sqrt{2}$, Fig. 5b, d, are $\pi$ out of phase relative to those along $(|\downarrow\rangle - i|\uparrow\rangle)/\sqrt{2}$, Fig. 5c, e. The overall measured entanglement fidelity is then $F \geq (F_1 + F_2)/2$. Hence, $F^{9T} = 75.86 \pm 4.3\%$ and $F^{5T} = 80.07 \pm 2.9\%$ for 9 T and 5 T, respectively. Our entanglement fidelity lower bound is comparable to the current state of the art at lower emission wavelengths (~900 nm)[18,19]. It could be further improved by improving the fidelity of spin rotations, using faster and more efficient detectors with lower jitter and dark counts, and improved filtering of the individual transitions.

## Discussion

In conclusion, our results realise the long-standing goal of spin-photon entanglement in a system capable of direct emission into the telecom C-band, a crucial building block for long-distance quantum networks. We have demonstrated high-fidelity optical spin initialization, coherent control, and projective measurement of an electron spin-qubit system resident in an InAs/InP QD. Furthermore, we measured the coherence time of an undisturbed single electron spin in a telecom-wavelength QD. Our results provide a stepping stone for a variety of quantum network applications such as long-distance quantum key distribution[35] and photonic quantum computing[36,37]. Many of these will in addition require indistinguishable photons as well as longer

coherence times. While the coherence properties of photons emitted from the QDs investigated here are promising[26], a full investigation of indistinguishability is still outstanding. The spin coherence time is similar to those measured in the more commonly used InAs/GaAs systems[16] and not quite ideal yet. Methods to improve it include applying electric fields to achieve a quieter environment[38,39], explore dynamical decoupling or nuclear spin engineering techniques[40,41], and ultimately using strain-free growth techniques and material systems[34]. Based on improved spin coherence, our entanglement protocol could be used for entanglement distribution, entangling distant QD spins[42]. This is particularly appealing given the low loss the telecom-wavelength photons will experience in optical fibres. Using the inherent coupling of the electron spin to the nuclear spin bath in an optimised sample could further give access to a local memory, with the outlook of using that in a quantum-memory-based quantum repeater scheme. Further, our system could be combined with photonic structures such as micropillars[18] and bullseye resonators[43,44] to allow for enhanced extraction efficiency. Such an efficient, coherent spin-photon interface is at the heart of multiphoton entanglement generation[5], itself a basic ingredient for all-photonic quantum repeaters[6]. To this end, selectivity would need to be added to the protocol used here. This could be done by switching to polarisation-encoded protocols[15,18,19], by using elastic scattering mechanisms[45,46], or

by using time-bin encoding in combination with a cavity structure inducing selective enhancement of predetermined transitions[16,17]. Here, the latter two have the additional benefit of being insensitive to nuclear magnetic field fluctuations and hence posing less restrictive conditions on the spin coherence time, making mulitphoton entanglement more accessible with our current system. In summary, our results show that InAs/InP QDs can host a spin system enabling direct entanglement with a telecom photon, and as such present versatile system for a range of entanglement-based quantum networking applications.

## Methods

### Sample description
Our sample consists of a single layer of self-assembled, droplet epitaxy InAs QDs, grown in an InP matrix in the centre of a planar cavity. The cavity is asymmetrical, with 20 bottom DBR (distributed Bragg reflector) pairs and three top DBR pairs. This enhances the efficiency of the structure by directing emission away from the substrate towards the collection optics. To further improve the photon collection efficiency, a 1-mm diameter, cubic zirconia solid immersion lens is attached to the top of the sample via 290 nm of HSQ.

### Experimental setup
The experimental setup utlised in this work allows us to implement above-band and resonant laser excitation as well as selective detection of resonance fluorescence from each of the four allowed transitions (see Supplementary Information 2). Ramsey interference experiments are conducted using a Ramsey interferometer where the time delay between the optical pulses can be adjusted via an optical delay line incorporated in one of the interferometer's arms. The fluorescence light is collected and filtered by a confocal microscope setup which removes the resonant laser contribution via cross-polarisation filtering. A free-space grating-based filter ($\Delta\lambda \sim 0.5$ nm or $\Delta\lambda \sim 0.12$ nm) is used to spatially separate individual excitonic complexes before detection.

### Synchronisation
In the experiments, the fixed repetition rate of our FFUltra rotation laser, 80 MHz, is used as a reference to synchronise the rest of the experimental apparatus. First, this reference signal seeds a phase-locked loop (LMX2595) from which a 40 MHZ sync output is obtained. This 40 MHz signal is distributed to Tektronix's AWG (AWG70000), Picoquant's LDH laser, and Hydrharp's time-tagging electronics.

## Data availability
The authors declare that the data supporting the findings of this study are available from the corresponding authors on request.

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

## Acknowledgements

The authors gratefully acknowledge the usage of wafer material developed during earlier projects in partnership with the National Epitaxy Facility at the University of Sheffield. They further acknowledge funding from the Ministry of Internal Affairs and Communications, Japan, via the project 'Research and Development for Construction of a Global Quantum Cryptography Network' in 'R&D of ICT Priority Technology' (JPMI00316). P.L. gratefully acknowledges funding from the Engineering and Physical Sciences Research Council (EPSRC) via the Centre for Doctoral Training in Connected Electronic and Photonic Systems, grant EP/S022139/1. The authors thank Andrea Barbiero and Ginny Shooter for their valuable discussions.

## Author contributions

A.J.S, R.M.S, D.A.R., and T.M. guided and supervised the project. J.S.-S. fabricated the emission-enhanced QD structure. P.L. and T.M. took the measurements and analysed the data. P.L. and T.M. wrote the manuscript. All authors discussed the results and commented on the manuscript.

## Competing interests

The authors declare no competing interests.
