## [Transparent Peer Review file · Nature Communications]

Spin-photon entanglement with direct photon emission in the telecom C-band

Corresponding Author: Mr Petros Laccotripes

Version 0:

Reviewer comments:

Reviewer #1

(Remarks to the Author)

The authors report on a spin-photon entanglement experiment using epitaxial quantum dots that emit directly in the telecom C-band. There is no doubt that the field of quantum communication would benefit greatly from qubits with direct optical interface in the telecom bands. Indeed, recent works on quantum dots emitting at shorter wavelength show that this system is particularly promising due to its large oscillator strength (and therefore possible rates) and sufficient spin-coherence for generating larger higher-order entangled photon states. The transfer of such techniques as spin-photon entanglement to quantum dots emitting in the telecom C-band is therefore an important milestone at the interface of the fields of quantum communication and solid-state nanoscale systems. The authors present a consistent study that goes beyond the state of the art, starting from spin initialization, over coherent control to the demonstration of spin-photon entanglement by measuring the classical and quantum correlations. The methods and data analysis are performed with rigor, and the authors present detailed supplemental data to support their results in the main text.

My opinion is therefore that this work is suitable for publication in Nature Communications. I am adding a few comments and questions to the authors:

1. "Wavelength conversion ... is inherently lossy and adds experimental complexity" In principle I agree with the authors, but nevertheless I believe this is a too simplified argument and should be discussed a bit more rigorously in the manuscript. E.g. it is not clear yet which spin lifetime/coherence can be achieved in quantum dots emitting at telecom bands. Therefore having a long-lived stationary qubit at shorter wavelength and adding frequency-conversion can still be a viable option in some cases.
2. "recent demonstration of ... Fourier-limited photon emission in the InAs/InP system [23]" I don't see that the data in the given reference supports this claim. The uncertainties are too big, and also the reported HOM measurements for the inelastically scattered photons do not support this. At most one should say here that the Fourier limit is "approached".
3. Can the authors comment on the data in Figure 3e for high time delays, why it doesn't seem to show nice oscillations and why it hasn't been included in the fit?
4. The authors say that the data in Figure 3c has been fitted with an exponentially decaying sinusoidal function. Can the authors comment on why the data does not start at zero and how this is included in the fit model?
5. So far the outlook mostly considers possible optimizations on the source. Can the authors give a bit more details in the outlook as to what the perspectives and limitations of their particular scheme for measuring spin-photon entanglement is and what the next steps would be to improve this or which protocols could be implemented in the next steps?

Reviewer #2

(Remarks to the Author)

In this work the authors demonstrate spin-photon entanglement using a InAs/InP quantum dot that emits in the telecom C band. The authors demonstrate initialization of the quantum dot electron spin, and demonstrate coherent control of the spin. They demonstrate entanglement between the spin state of the electron and the frequency of the emitted photon with a fidelity

of 80%. These techniques have previously been demonstrated in other quantum dot systems, but are shown here for a quantum dot that emits in the telecom C band for the first time.

This is interesting work, as spin-photon entanglement is a key step towards entanglement distribution across quantum networks and for the generation of multi-photon states, and operation in the telecom C band is key for large-scale quantum networks. For that reason, translating well-established spin control techniques to this new wavelength is an important and interesting step.

However, there are several important details missing from the paper, and these need to be addressed.

(1) Many of the experimentally measured values given in the paper do not have an error. Most notably is the abstract where a fidelity is given with no error, which is surprising given the following sentence 'more than 10 standard deviations above the classical limit'. In this case it seems that it is a simple omission since this value is presented later in the paper with an error. However there are several other values such as the fidelity of spin preparation, the fidelity of the $\pi/2$ pulse, and the extracted inhomogeneous dephasing time all of which do not have errors.

(2) For the data in Figure 5 the error bars are very large compared to the amplitude of the oscillation. What is the error of the amplitude and offset of the fit? Why is the amplitude of oscillations lower for the data in Fig 5b compared to 5c? How does this affect the extracted fidelity and error? Also, how does the (imperfect) fidelity of the $\pi/2$ or $3\pi/2$ pulse affect the extracted fidelity of spin-photon entanglement?

(3) What transition is the spin rotation pulse interacting with? On page 7 it says that a large detuning is used, but does not state detuning from which transition. It would be helpful to include this on an energy level diagram in Figure 3.

(4) What is the value of magnetic field used for the data in Figure 3e/f? And does the measured frequency of the oscillations make sense for this value of B?

(5) The timing sequence for the data presented in Figure 4 is not clear. At what time does the entanglement pulse arrive? From the data in figure 4d it seems to be at approximately 1ns? This is important given the discussion of the integration windows of either [1ns,4ns] and [1.14ns,1.7ns]. If the excitation pulse with a duration of 300ps arrives at approximately 1ns then I understand why neglecting the early part of the photon emission reduces the g_2 , but why remove the photons from 1.7ns to 4ns? This cannot be due to residual laser contribution. Also, there is no discussion on the impact of photon brightness by choosing this short time window.

(6) There is no discussion of photon extraction efficiency for the quantum dot system. Whilst I appreciate this is not the focus of this paper, it would still be useful to include an indication.

(7) There are a few places where the figures are not completely clear, or the figure captions are missing details:

(i) In the experimental setup diagram in Fig 1b, the beam path through the beam splitter and translation stage at the bottom does not make sense, I think the beam should exit the upper right of the beam splitter?

(ii) Do the blue data points in Fig 2c have error bars?

(iii) In Figure 3 it should be made clearer (either in the figure or in the caption) that for the data in 3c and 3d there is only one rotation pulse, and in 3e and 3f there are two pulses

(iv) In Fig 4d/e the choice of colour for the two lines for the red frequency photons and the blue frequency photons are too similar. Also, the green and purple dashed lines (the integration windows) are not labelled/described in the caption.

(v) The black and coloured sinusoidal lines in Figure 5 are not labelled. I presume the coloured lines are after the detector deconvolution?

I would like the authors to address the above points, as at this stage I cannot give a recommendation.

Reviewer #3

(Remarks to the Author)

The manuscript from Laccotripes et al, reports measurements of spin-photon entanglement with InAs quantum dots in InP. Spin-photon entanglement has been demonstrated in InAs/GaAs QDs, but here it has been achieved with C-band emitters, which potentially enables long-distance communication. It is therefore a rather important step towards important applications such as QKD and quantum computing. The methodology, data analysis and interpretation are sound, with very detailed supporting material.

I think the manuscript should be accepted for publication after addressing the following comments:

- A known drawback for C-band emitters is the low indistinguishability of the emitted photon, which is not reported in this work. Is this issue resolved? The authors mention Fourier-limited photon emission (ref 23, available as preprint) but the work only concerns CW operation and two-photon interference measurement. I recommend adding a few comments on the performance of the photon emitters in view of the envisioned applications requiring indistinguishable photons (e.g. computing and communication).

- Also, towards applications such as quantum computing, is the frequency entanglement shown here sufficient, or would other schemes be necessary, e.g. path or time-bin encoding? Given the broad readership of Nature Communications, I suggest providing more info on the challenges ahead.

- Can the author comment on the level of fidelity achieved in entanglement and what are the main limitations?
- Caption of Fig. 3d (and text) refers to a single de-tuning, but three power dependencies as a function of de-tuning are shown. The explanation for the sub-linear dependence is very technical, maybe it can be rewritten for broader audiences, following the same level of clarity used in the rest of the manuscript.

Version 1:

Reviewer comments:

Reviewer #2

(Remarks to the Author)

Thank you to the authors for addressing my comments, and those of the other reviewers. I think the changes that the authors have made to the manuscript have improved its quality and clarity. I recommend it for publication in Nature Communications.

Reviewer #3

(Remarks to the Author)

The authors have addressed all my comments, therefore I support that the work is published in Nature Communications.

Answers to the referee's comments

We would like to thank all referees for their careful review of our work and for their time and effort spent on it. We sincerely appreciate their valuable comments and suggestions, which have significantly helped us improve the quality of our manuscript. Below, we address their comments individually, with reviewer comments given in **bold**.

Response to comments from Reviewer 1:

The authors report on a spin-photon entanglement experiment using epitaxial quantum dots that emit directly in the telecom C-band. There is no doubt that the field of quantum communication would benefit greatly from qubits with direct optical interface in the telecom bands. Indeed, recent works on quantum dots emitting at shorter wavelength show that this system is particularly promising due to its large oscillator strength (and therefore possible rates) and sufficient spin-coherence for generating larger higher-order entangled photon states. The transfer of such techniques as spin-photon entanglement to quantum dots emitting in the telecom C-band is therefore an important milestone at the interface of the fields of quantum communication and solid-state nanoscale systems. The authors present a consistent study that goes beyond the state of the art, starting from spin initialization, over coherent control to the demonstration of spin-photon entanglement by measuring the classical and quantum correlations. The methods and data analysis are performed with rigor, and the authors present detailed supplemental data to support their results in the main text.

My opinion is therefore that this work is suitable for publication in Nature Communications. I am adding a few comments and questions to the authors:

We thank the reviewer for their very positive assessment of the impact and quality of our work. In the following, we endeavour to fully address their comments.

1. "Wavelength conversion ... is inherently lossy and adds experimental complexity" In principle I agree with the authors, but nevertheless I believe this is a too simplified argument and should be discussed a bit more rigorously in the manuscript. E.g. it is not clear yet which spin lifetime/coherence can be achieved in quantum dots emitting at telecom bands. Therefore having a long-lived stationary qubit at shorter wavelength and adding frequency-conversion can still be a viable option in some cases.

Our response:

We understand and acknowledge the benefits of wavelength conversion. In the revised manuscript, we have expanded our discussion to address the nuances of this topic more rigorously. Specifically, we recognize that applications requiring long coherence times, such as quantum memories, can indeed benefit from wavelength conversion. This is particularly pertinent given the limited coherence times currently observed in telecom wavelength quantum dots. However, the added experimental complexity and inherent additional attenuation can considerably impact the efficiency of entanglement distribution and limit the feasibility and performance of cluster state generation. We have added the following in the revised manuscript:

"Wavelength conversion is one option to circumvent this problem. While it can be beneficial for applications requiring long coherence times, such as quantum memories [*Radnaev, A. et al. Nature Physics 6.11 (2010): 894-899*], the increased experimental complexity and inherent

losses can significantly hinder the efficiency of entanglement distribution and render applications such as multiphoton entangled state generation unfeasible”.

2. "recent demonstration of ... Fourier-limited photon emission in the InAs/InP system [23]" I don't see that the data in the given reference supports this claim. The uncertainties are too big, and also the reported HOM measurements for the inelastically scattered photons do not support this. At most one should say here that the Fourier limit is "approached".

Our response:

We thank the reviewer for their suggestion. We have altered the manuscript accordingly,

“Most prominently, impressive progress has been made using quantum dots (QDs) based on strain-relaxed InAs/GaAs or InAs/InP, culminating in the recent demonstrations of coherent spin manipulation in the strain-relaxed InAs/GaAs system, and **photon emission approaching the Fourier limit** in the InAs/InP system, respectively.”

3. Can the authors comment on the data in Figure 3e for high time delays, why it doesn't seem to show nice oscillations and why it hasn't been included in the fit?

Our response:

We thank the reviewer for their comment regarding the data in Figure 3e. For long ODL delays we see an increase in the noise in our data and a total collapse of oscillations which is not consistent with an exponential decay. The source of this collapse at long delays in our Michelson interferometer is currently unclear and will require further investigation. It is possible that this is due to a technical limitation in our experimental setup but we believe it is strongly affected by dynamical nuclear spin polarisation which makes it difficult to extract the T_2^* coherence time for long time delays [*Sun, Z. et al. Physical Review B 93.24 (2016): 241302*]. Therefore, to ensure the accuracy and reliability of our results, we have excluded the high-delay data from the fit. In response to your comment, we have added the above information to the supplementary information, Section V.

4. The authors say that the data in Figure 3c has been fitted with an exponentially decaying sinusoidal function. Can the authors comment on why the data does not start at zero and how this is included in the fit model?

Our response:

We thank the reviewer for their comment regarding the data in Figure 3c and the fitting model used. We attribute this effect to imperfect spin initialisation. Specifically, the pumping time is fixed at 6 ns, resulting to a spin initialisation fidelity of approximately 93% as seen in Figure 2c. This is included in the fitting function as a background offset parameter. To make this clearer to the reader, we have incorporated a new sentence in the manuscript.

“Moreover, due to the finite fidelity of spin initialisation (~93 % for 6 ns of optical pumping), the integrated emission counts do not start at zero in the limit of 0 μ W rotation pulse power.”

5. So far the outlook mostly considers possible optimizations on the source. Can the

authors give a bit more details in the outlook as to what the perspectives and limitations of their particular scheme for measuring spin-photon entanglement is and what the next steps would be to improve this or which protocols could be implemented in the next steps?

Our response:

We thank the reviewer for pointing this out. Indeed, there are a number of applications within reach from our proof-of-principle demonstration. For example, with improved spin coherence times, our entanglement protocol could be used for entanglement distribution, entangling distant quantum dot spins. This is particularly appealing given the low loss the telecom-wavelength photons will experience in optical fibres. Using the inherent coupling of the electron spin to the nuclear spin bath in an optimised sample could further give access to a local memory, with the outlook of using that in a quantum-memory based quantum repeater scheme.

Another interesting direction would be the generation of multiphoton entangled states. These states are very interesting in the context of all-photon quantum repeaters. Certain additions to our proof-of-principle system will be necessary. Foremost, the spin coherence will need to be improved to extend to at least one repetition cycle of a protocol with inherent spin rephasing (such as the Lodahl group's time-bin encoded protocol). Further, cycling transitions need to be introduced, either by selectively enhancing the vertical transitions in a cavity, or by switching to a polarisation encoded protocol as used by the Gershoni's and Senellart's groups. All protocols will further benefit from enhanced extraction efficiencies in appropriate photonic structures.

We have included the following the manuscript in response to your comments.

“Based on improved spin coherence, our entanglement protocol could be used for entanglement distribution, entangling distant quantum dot spins. This is particularly appealing given the low loss the telecom-wavelength photons will experience in optical fibres. Using the inherent coupling of the electron spin to the nuclear spin bath in an optimised sample could further give access to a local memory, with the outlook of using that in a quantum-memory based quantum repeater scheme. Further, our system could be combined with photonic structures such as micropillars and bullseye resonators to allow for enhanced extraction efficiency.

Such an efficient, coherent spin-photon interface is at the heart of multiphoton entanglement generation, itself a basic ingredient for all-photon quantum repeaters. To this end, selectivity would need to be added to the protocol used here. This could be done by switching to polarisation encoded protocols, by using elastic scattering mechanisms, or by using time-bin encoding in combination with a cavity structure inducing selective enhancement of predetermined transitions. Here, the latter two have the additional benefit of being insensitive to nuclear magnetic field fluctuations and hence posing less restrictive conditions on the spin coherence time, making multiphoton entanglement more accessible with our current system.

In summary, our results show that InAs/InP QDs can host a spin system enabling direct entanglement with a telecom photon, and as such present versatile system for a range of entanglement-based quantum networking applications.”

Response to comments from Reviewer 2:

In this work the authors demonstrate spin-photon entanglement using a InAs/InP quantum dot that emits in the telecom C band. The authors demonstrate initialisation of the quantum dot electron spin, and demonstrate coherent control of the spin. They demonstrate entanglement between the spin state of the electron and the frequency of the emitted photon with a fidelity of 80%. These techniques have previously been

demonstrated in other quantum dot systems, but are shown here for a quantum dot that emits in the telecom C band for the first time.

This is interesting work, as spin-photon entanglement is a key step towards entanglement distribution across quantum networks and for the generation of multi-photon states, and operation in the telecom C band is key for large-scale quantum networks. For that reason, translating well-established spin control techniques to this new wavelength is an important and interesting step.

However, there are several important details missing from the paper, and these need to be addressed.

We thank the reviewer for their positive assessment of our work and its impact. In the following, we will address their remaining comments and concerns.

1. Many of the experimentally measured values given in the paper do not have an error. Most notably is the abstract where a fidelity is given with no error, which is surprising given the following sentence ‘more than 10 standard deviations above the classical limit’. In this case it seems that it is a simple omission since this value is presented later in the paper with an error. However there are several other values such as the fidelity of spin preparation, the fidelity of the $\pi/2$ pulse, and the extracted inhomogeneous dephasing time all of which do not have errors.

Our response:

We thank the reviewer for pointing out this issue. We’re sorry for the omission. We have added all error values in the manuscript.

2. For the data in Figure 5 the error bars are very large compared to the amplitude of the oscillation. What is the error of the amplitude and offset of the fit? Why is the amplitude of oscillations lower for the data in Fig 5b compared to 5c? How does this affect the extracted fidelity and error? Also, how does the (imperfect) fidelity of the $\pi/2$ or $3\pi/2$ pulse affect the extracted fidelity of spin-photon entanglement?

Our response:

We thank the reviewer for this valuable comment. The error bars on the raw data are determined by the number of coincidences detected during the measurement time. This is a combination of acquisition time as well as efficiency of the detection setup, and was smaller for 9 T data compared to 5 T.

As the reviewer has pointed out, the visibility of the oscillations is lower for the data present in 5b compared to 5c. We suspect that this stems from an imperfect $\pi/2$ pulse which resulted in the electron spin not experiencing a true $\pi/2$ rotation thus resulting in lower visibility for the oscillations. For each dataset (each panel in figure 5) we measured, we performed individual power calibrations. However, the effective rotation angle is very sensitive to the laser power experienced by the quantum dot, and hence even a small misalignment of the microscope can affect the spin rotation fidelity. Further, the two interfering decay paths have orthogonal polarisation. If one of these is detected more efficiently, the measured amplitude also decreases. While we make every attempt to align our detection polarisation with an equal superposition of both polarisations, variations in alignment can still occur. A lower visibility amplitude will lead directly to a degradation in the entanglement fidelity measured. We have

added the visibility values along with their respective errors in the supplementary material. The offset has no error because the data have been normalised to 0.5.

We have included the following in the revised manuscript in response to the reviewer's comment:

“While clear oscillations are visible in the raw data, the period of 66 ps at 9 T approaches the timing jitter of our detector system (40 ps) and hence oscillation amplitudes are reduced. Further technical limitations include variations in the effective rotation laser power due to drifts in the microscope alignment over the measurement duration, as well as imperfect calibration of the superposition detection basis. We suspect these limitations to be the root cause for the difference in the oscillation amplitude between Figure 5(b) and (c).”

3. What transition is the spin rotation pulse interacting with? On page 7 it says that a large detuning is used, but does not state detuning from which transition. It would be helpful to include this on an energy level diagram in Figure 3.

Our response:

We thank the reviewer for pointing this out. The rotation pulse is red-detuned from the lowest energy transition.

We have altered the caption of figure 3c to reflect the above:

“Integrated emission intensity as a function of the rotation pulse power for three different red-detunings from the lowest energy transition, demonstrating coherent control of the electron's spin.”

The main text has been altered as follows:

“We employ a circularly polarised pulse to ensure that the probability amplitudes of the two transitions in the effective Λ system add constructively, and large red-detunings, from the lowest energy transition, to minimise undesired incoherent excitation”.

We have added the following text and figure to Section V of the supplementary to explain the rotation process.

“For single spin rotations, a broadband high-intensity laser pulse can induce a Stimulated Raman adiabatic Passage (STIRAP). The effective field experience by the QD can be much larger than the applied magnetic field, resulting in an effective Rabi frequency $\Omega_{eff} =$

$\sqrt{\frac{\Delta^2|\Omega_H|^2|\Omega_V|^2}{(4\Delta^2+\Gamma^2)^2}} + \delta^2 \approx |\Omega_H||\Omega_V|/2\Delta$, where Ω_j is the Rabi frequency of the j-polarised transition, δ is the Larmor frequency of the ground state and Δ is the detuning of the rotation pulse.”

4. What is the value of magnetic field used for the data in Figure 3e/f? And does the measured frequency of the oscillations make sense for this value of B?

Our response:

We thank the reviewer for pointing this out. The data present in the figure are at 5T for a ground state Larmor precession of $T_e = 117.7 \pm 1.4$ ps. A very similar oscillation period is observed in the Ramsey fringes ($T_{Ramsey} = 120 \pm 1$ ps).

We incorporated the following in the main manuscript:

Rabi oscillations demonstrate the rotation of a qubit by a single axis, U(1) control. Full coherent control, SU(2) control, is achieved by rotation about a second axis, ϕ , which can be realised by utilising the inherent Larmor precession in the applied magnetic field. Here, we chose a magnetic field of 5 T for a Larmor precession of ~ 120 ps. We employ detuned circularly polarised ps-pulses and make use of the precession to achieve rotation by θ and ϕ , respectively.

5. The timing sequence for the data presented in Figure 4 is not clear. At what time does the entanglement pulse arrive? From the data in figure 4d it seems to be at approximately 1ns? This is important given the discussion of the integration windows of either [1ns,4ns] and [1.14ns,1.7ns]. If the excitation pulse with a duration of 300ps arrives at approximately 1ns then I understand why neglecting the early part of the photon emission reduces the g_2 , but why remove the photons from 1.7ns to 4ns? This cannot be due to residual laser contribution. Also, there is no discussion on the impact of photon brightness by choosing this short time window.

Our response:

We thank the reviewer for this comment. As the reviewer pointed out the entanglement pulse arrives at approximately 800 ps. We choose an evaluation window after the application of the pulse. By filtering in time, we avoid any contributions from the scattered laser photons. Moreover, we omit part of the tail because eventually, counts from the quantum dot are competing with detector noise.

We have added this information to the manuscript as follows:
 ‘The excitation pulse arrives at approximately 800 ps with a duration of 300 ps. By time-filtering we avoid any contributions from the scattered laser photons, as well as the detector’s dark count rates.’

6. There is no discussion of photon extraction efficiency for the quantum dot system. Whilst I appreciate this is not the focus of this paper, it would still be useful to include an indication.

Our response:

We thank the reviewer for their comment. Indeed, photon extraction efficiency is not the focus of this paper, and is notoriously hard to quantify. In the following, we attempt to nevertheless provide an estimate. We have simulated the first-lens brightness of our system using a 1D-transfer matrix approach according to Ma *et al.* [Journal of Applied Physics 115 023106 (2-14)]. This lets us estimate a first-lens brightness of around 15%. The overall detection efficiency is a further a composite of the following factors, which we estimate from the experimental setup:

Efficiency	Origin
0.15	First-lens brightness from simulations
0.5	Polarisation rejection
0.5	Fibre coupling efficiency
0.4	Grating filter efficiency
0.5	Additional losses in detection setup, incl. detector efficiency
0.0075	Total collection efficiency

However, collection efficiency is not the only factor reducing our count rates. From our spectra under non-resonant excitation, we expect the X- to be created less than 40% of the time. In

addition, excitation efficiency and quantum efficiency will both affect the measured count rate, but are harder to quantify.

The above paragraph has been added to the supplementary information.

7. There are a few places where the figures are not completely clear, or the figure captions are missing details:

Our response:

We thank the reviewer for pointing these out. We apologise for the omissions. We have addressed all points individually.

(i) In the experimental setup diagram in Fig 1b, the beam path through the beam splitter and translation stage at the bottom does not make sense, I think the beam should exit the upper right of the beam splitter?

The beam path in Figure 1b has been corrected.

(ii) Do the blue data points in Fig 2c have error bars?

Error bars have been added but are within the square data markers.

(iii) In Figure 3 it should be made clearer (either in the figure or in the caption) that for the data in 3c and 3d there is only one rotation pulse, and in 3e and 3f there are two pulses

We have altered the caption:

“(c) Integrated emission intensity as a function of a single rotation pulse’s power for three different red-detunings from the lowest energy transition, demonstrating coherent control of the electron’s spin.”

“(e) Ramsey interference for a pair of rotation pulses...”

(iv) In Fig 4d/e the choice of colour for the two lines for the red frequency photons and the blue frequency photons are too similar. Also, the green and purple dashed lines (the integration windows) are not labelled/described in the caption.

We have updated the colour choice in the figure to make the colours stand out more. We have also added the following to the caption of the figure to describe the two integration windows:

“(c) Second-order autocorrelation measurement to evaluate the purity of our source. Two different time windows, [1.15 ns, 1.7 ns] marked in green, and [1 ns, 4 ns] marked in purple, in panels (d) and (e), are considered which correspond to part or all of entanglement pulse decay, respectively.

(v) The black and coloured sinusoidal lines in Figure 5 are not labelled. I presume the coloured lines are after the detector deconvolution?

Yes, this is correct. We have added the following to the caption:

“(a) Schematic of the pulse sequence used for the correlation measurements in superposition bases. (b) and (c) Time-resolved coincidence events when the spin state was projected onto $|\uparrow\rangle + i|\downarrow\rangle$ and $|\uparrow\rangle - i|\downarrow\rangle$, respectively, recorded at a magnetic field of 9 T. (d) and (e) identical

measurements to (b) and (c), recorded at a magnetic field of 5 T. Black lines correspond to fits of the original data and coloured lines are after deconvolving with the detectors' jitter."

Response to comments from Reviewer 3:

The manuscript from Laccotripes et al, reports measurements of spin-photon entanglement with InAs quantum dots in InP. Spin-photon entanglement has been demonstrated in InAs/GaAs QDs, but here it has been achieved with C-band emitters, which potentially enables long-distance communication. It is therefore a rather important step towards important applications such as QKD and quantum computing. The methodology, data analysis and interpretation are sound, with very detailed supporting material.

I think the manuscript should be accepted for publication after addressing the following comments:

We appreciate the reviewer's positive evaluation of our work and its significance. In the following, we address any remaining comments and concerns.

1. A known drawback for C-band emitters is the low indistinguishability of the emitted photon, which is not reported in this work. Is this issue resolved? The authors mention Fourier-limited photon emission (ref 23, available as preprint) but the work only concerns CW operation and two-photon interference measurement. I recommend adding a few comments on the performance of the photon emitters in view of the envisioned applications requiring indistinguishable photons (e.g. computing and communication).

Our response:

We thank the reviewer for their comment regarding the indistinguishability of the emitted photons. We agree that indistinguishability is an important metric for many applications in quantum networking, where we see the main benefit of the native telecom wavelength emission. As the reviewer mentions, the coherence properties of photons emitted from our dots are very promising for good indistinguishability. However, we are still working on pulsed HOM measurements to determine the exact value of indistinguishability, and we cannot report it just yet. Nevertheless we have added the viewpoint of indistinguishability to the new discussion of envisioned applications for our quantum dots in the conclusion of the paper.

2. Also, towards applications such as quantum computing, is the frequency entanglement shown here sufficient, or would other schemes be necessary, e.g. path or time-bin encoding? Given the broad readership of Nature Communications, I suggest providing more info on the challenges ahead.

Our response:

We thank the reviewer for pointing this out. Reviewer 1 had a very similar comment, which we have addressed in detail. We would therefore like to refer the reviewer to our answer given above.

3. Can the author comment on the level of fidelity achieved in entanglement and what are the main limitations?

Our response:

The spin photon entanglement fidelity of 80.07(2.9) % achieved in this work is the only ever reported from a quantum dot emitting directly in the telecom C-band. The latest works that utilise quantum dots and polarisation encoding are by D. Gershoni's and P. Sennelart's teams with dots emitting around 900 nm and have achieved very high spin-photon entanglement fidelities of 90(1) % and 67(1) %, respectively. Using time-bin encoding P. Lodahl's group have demonstrated a 2-qubit fidelity of 76.3(5) % with a similar emission wavelength as the other works. These numbers show that we achieve fidelities comparable to the current state of the art.

The following factors limit the achievable entanglement fidelity in our case:

- Imperfect spin rotations caused by power drifts in the setup.
- Inherent trade-off between detecting high Larmor frequencies with the finite timing resolution of our detectors versus detecting single closely spaced transitions in the computational basis, as explained in the Supplementary information Section VI.
- Detector dark counts.

We have summarised these for convenience at the end of the fidelity discussion, Section IV, in the main text.

“Our fidelity lower bound is comparable to the current state of the art at lower emission wavelengths (~ 900 nm) [18, 19]. It could be further improved by improving the fidelity of spin rotations, using faster and more efficient detectors with lower jitter and dark counts, and improved filtering of the individual transitions.”

4. Caption of Fig. 3d (and text) refers to a single de-tuning, but three power dependencies as a function of de-tuning are shown. The explanation for the sub-linear dependence is very technical, maybe it can be rewritten for broader audiences, following the same level of clarity used in the rest of the manuscript.

Our response:

We thank the reviewer for their suggestion. We have altered the caption of figure 3d accordingly:

“Rotation angle versus pulse power fit results in an average power dependence of $\theta = \alpha P_{rot}^{0.77}$ for the three detunings.”

We have also altered the text in the main manuscript as follows:

‘While a linear dependence is expected for the ideal STIRAP protocol, the sub-linear dependence observed here is an expression of the breakdown of the adiabatic elimination of the excited states in the rotation process. This means that there is a finite probability of finding the system in the excited state, from which radiative decay can take place.’